# The Reception of Jane Austen in Early Modern China:
# A Canonical Perspective

Helong Zhang

Institute of Literary Studies, Shanghai International Studies University, Shanghai 200083, China; 1336@shisu.edu.cn

**Abstract:** In China, Jane Austen has undergone an amazing metamorphosis from an obscure foreign writer disregarded or disapproved of for a long period to a great novelist highly acclaimed and fully acknowledged. Only recent years have seen the publication of a few scholarly articles on the reception trajectories of Austen in the Chinese academic world. This article revisits the issue, particularly the reception of Austen in early modern China from a canonical perspective. During the first major wave of literary translation, Austen was absent in the translation projects of dominant male translators, especially in Lin Shu's choice. It was not because of their gender discrimination as generally considered, but because of their lack of canon consciousness. The literary light of Austen, too bright and too sparkling to ignore, was finally shed upon the Chinese land, but her canonical place was not instantly recognized. The wartime translators' efforts to render *Pride and Prejudice* into Chinese reflect the difficulty in the making of a canonical Austen under very different historical circumstances.

**Keywords:** reception; modern China; Jane Austen; *Pride and Prejudice*; canon consciousness





## 1. Introduction

In China, Jane Austen has attained her canonical place over the past three decades, though she had been available to Chinese readers for more than one hundred years. It is well worth exploring how and why such an English literary master has undergone such an amazing metamorphosis from obscure foreign writer, long disregarded or disapproved of by Chinese translators and critics, to a great novelist highly acclaimed and fully acknowledged. Despite the fact that there has been considerable research interest in the reception of foreign literature for several decades, (Zhang 2015, pp. 1–2) only in recent years has there been publication of a few scholarly articles on the reception trajectories of Austen in the Chinese academic world.[1] A large number of foreign writers (including many British writers) were considered 'classic' or 'world-famous' as soon as they were introduced into China, but it proves otherwise in the case of Austen. China might have been the 'missing link' for which the 'global' Austen seemed to have fallen short throughout much of the twentieth century.

In 'Jane Austen's One Hundred Years in China,' I briefly discussed the social, historical and cultural milieu in which the reception of a foreign writer like Austen might have been greatly influenced. In this article, I would like to take my understanding a little further in a different light by revisiting the issue, particularly the reception of Austen in early modern China. As it is, the canonization of a writer in his or her home country would not have been the same as in another country. People in different countries usually have different literary tastes, aesthetic habits, critical standards, moral values, social mores, historical experiences, and, in particular, different ideological needs as far as the translation choice is concerned. The cross-cultural and critical influences across countries play special roles during the process of canonizing a foreign writer. Arguably, the way Chinese intellectuals

regarded Austen in the early period would have to be measured in connection with how much she had been critically acclaimed in the Anglo-American academic world.

Austen, long ignored or untranslated in early modern China, had enjoyed much less popularity than many other British authors among Chinese readers, and her literary image remained relatively vague to many Chinese intellectuals. This article will explore why Austen was absent in the first major wave of literary translation in early modern China, and how her bright and sparkling light was finally shed upon the land of China. In addition, the author will go on to discuss how *Pride and Prejudice* was translated into Chinese for the first time and why the wartime translators did not create successful or influential Chinese versions in the making of a canonical Austen under very different historical circumstances.

## 2. The Absence of Austen

Since the high tide of literary translation emerged at the end of the nineteenth century, a large number of British writers have been rendered to Chinese readers and have greatly refreshed their literary imagination. Many of these, such as William Shakespeare, John Milton, Geoffrey Chaucer, Percy Bysshe Shelley, George Gordon Byron, or Charles Dickens, were indisputably treated as 'classic' or 'greatest' as soon as their names showed up for the first time in Chinese translation. In the first two decades of the twentieth century, Lin Shu (1852–1924), the foremost translator of his time, had offered to Chinese readers five plays by Shakespeare, Daniel Defoe's *Robinson Crusoe*, Jonathan Swift's *Gulliver's Travels*, two novels by Henry Fielding, three novels by Walter Scott, five novels by Charles Dickens, twenty-five novels by Henry Rider Haggard, seven novels by Arthur Conan Doyle, one novel by Robert Louis Stevenson, and works of many minor or unknown British writers (Yu 2010, pp. 348–71), except Austen's. It is well-known that Lin Shu did not know any foreign language but wrote very well in classical Chinese. He chose to produce his literary translations in collaboration with his oral interpreters. Many literary works he translated 'reached readers throughout China and influenced a generation of writers who grew up reading those translations' (Wang 2017, p. 174).

The male Chinese translators played dominant roles in the first wave of literary translation in modern China. Critics tend to believe that Austen was excluded from their translation projects in the early decades mainly because they held negative views of female authors. Another implausible allegation is that Lin Shu personally evinced no special interest in Austen on account of his 'conservatism,' which can be confirmed by his 'reactionary' attitudes towards the New Culture Movement when he engaged in fierce debates with the radical intelligentsia. What has been overlooked by critics in fact is that Lin Shu did translate quite a few Anglo-American female authors after he had achieved tremendous success by translating *Uncle Tom's Cabin* (an anti-slavery novel by the nineteenth-century American woman writer Harriet Beecher Stowe) into Chinese in 1901. His *oeuvre* is a complex body of translated works, including Baroness Emma Orczy's *The Scarlet Pimpernel*, Florence L. Barclay's *The Rosary*, Mary Cowden Clarke's *Tale II: The Thane's Daughter* (taken from *The Girlhood of Shakespeare's Heroines*), Sophia H. Maclehose's *Tales from Spencer Chosen from the Faerie Queene*, Emma Southworth's *The Changed Brides*, etc. Lin Shu's gender discrimination was not as conspicuous as generally considered if we refer to his introductory remarks in a postscript to *The Black Slaves' Plea to Heaven* (the translated title of Mrs. Stowe's novel *Uncle Tom's Cabin*). He clearly displayed open-mindedness and ready acceptance of the newly emerged gender issues by saying that 'both man and woman enjoy equal importance in Western countries.' (Lin 2010a, p. 91). His translation of Mrs. Stowe's novel is concrete evidence that he did not cherish negative attitudes towards gender issues.

The reason Austen was disregarded is probably that Lin Shu and many other early translators would prefer authors who were 'either long-established in English literature or had just achieved great success at the moment of their translation' (Zhang 2011, p. 104). In other words, a foreign writer who had already achieved a canonical place in their home country, or a bestselling author who was apparently blessed with high commercial values,

would most likely receive definite preference from translators and especially publishers. Sun Shuo, an Austen scholar in China, says, 'Austen had gained both fame and popularity by the early twentieth-century, and therefore the reason that Lin and his collaborators did not translate her works is unlikely to have been that she was unknown.' (Sun 2020, p. 4). Austen was certainly not an obscure novelist in the 1910s and 1920s. However, this does not mean that she was as renowned or deeply influential as many long-established British authors. Neither was she a contemporary bestselling novelist during Lin Shu's glorious period of translation.

Austen had not been fully subsumed in the English literary canon up to the turn of the twentieth century. It is believed that 'not until the 1920s had English literary scholars and critics reached a consensus on the significance of Austen as a great novelist.' (Gong 2018, p. 42). Controversies arose repeatedly in the nineteenth and early twentieth century in the English literary world. The American critic A. Walton Litz noted that an 'imposing' group of 'anti-Janeites', composed of Mark Twain, Henry James, Charlotte Brontë, D. H. Lawrence and Kingsley Amis, depreciated or felt repelled by her novels. Litz claimed that 'in almost every case the adverse judgement merely reveals the special limitations or eccentricities of the critic, leaving Austen relatively untouched' (Litz 1975, p. 672). However, 'the adverse judgement' had been a major factor that obviously hindered the reception of her works in the beginning of the twentieth century in China. An author, whose canonical place had not been fully recognized in English literary history, would not have attracted attention from a translator or publisher unless an extremely farsighted Chinese critic could remove their doubts convincingly and make themselves understood. As Virginia Woolf said of Austen, 'Of all great writers she is the most difficult to catch in the act of greatness' (Qtd in Southam 1987, p. 301).

To publishers, a contemporary bestselling author is the most likely to offer commercial success or best reception in the market of literary translation. Austen had not been a bestselling novelist throughout most of the nineteenth century. As Bruce Stoval asserts, 'Until the publication of the *Memoir* in 1870, Jane Austen's novels had a small, if growing and appreciative, audience' (Stovel 1997, p. 232). New editions of her novels flooded the market after 1870. However, their commercial values could not be mentioned in the same breath as those English bestsellers in Lin Shu's time. For example, the reason why Lin Shu and his collaborator had a predilection for *The Rosary* was plainly evident: the novel was an immediate success after its publication in 1909 and its author, Florence L. Barclay, was one of the most popular British novelists of Lin Shu's time. According to the *New York Times*, the novel was the No.1 bestselling novel of 1910 in the United States. For another example, Baroness Emma Orczy's *The Scarlet Pimpernel* was translated into Chinese in 1905, the same year as its English publication, because it was an immediate success too. The author's play of the same title, after which the novel was written, had been one of the most popular shows for several years in London and eventually more than 2000 performances were shown in Britain since its opening at the New Theatre on 5 January 1905.

It is therefore not justifiable to accuse Lin Shu and other translators of disregarding female authors. Lin Shu and his collaborators made an overwhelming choice of male authors for the reason that the greatest proportion of writers who had been established or canonized in English literary history were male. In William J. Long's *English Literature*, an extremely popular textbook first published in 1909, Austen was classified as one of the 'secondary writers of Romanticism' in contrast to Wordsworth, Coleridge, Southey, Byron, Shelley, Keats, Scott, Lamb, and De Quincey. Long argued that the age of the French Revolution also 'produced a large number of minor writers, who followed more or less closely the example of its great leaders. Among novelists we have Jane Austen, Frances Burney, Maria Edgeworth, Jane Porter, and Susan Ferrier—all women, be it noted' (Long 1909, p. 436). Obviously, if Lin Shu had displayed gender bias in his translation choices, it would not have been his or his collaborators' fault in a large sense.

In addition, there had been more male bestselling authors than female ones. As a result, Lin Shu had accomplished an amazing feat by translating twenty-five of Haggard's novels

into Chinese. These bestselling works would predictably serve the needs of the domestic literary market. *Joan Haste*, one of Haggard's bestsellers, enjoyed great popularity with Chinese readers. Based on his translation, the Comprehensive Drama School in Shanghai staged a play in March 1908, and the staging 'established a framework for performance of Chinese 'spoken drama' (Tam 2019, p. 24). For a moment, Lin Shu had readily recognized Haggard as great a living author as Walter Scott. He says in a preface to his 1905 Chinese version of *Ivanhoe*, 'Among the greatest writers, I know Dumas the father and the son in France; I know Walter Scott and Haggard in Britain' (Lin 2010b, p. 104). Obviously, he was misguided by a simple formula that popularity equals greatness.

Considering the historical circumstances under which the high tide of literary translation occurred, 'conservative' Chinese intellectuals as well as radical ones, who had repeatedly experienced or witnessed foreign invasions, wars and other civil disturbances in modern China, were all patriots. They would show great interest in 'novels that could help to reform society, strengthen a weak country, encourage the patriotic passion of Chinese reader, or simply serve the commercial purpose of publishers' (Zhang 2011, p. 105). Austen's novels, which represented a restricted world—'3 or 4 families in a country village'—were far from fulfilling their driving ambitions to transform China into a powerful modern state or to mobilize Chinese readers to rejuvenate China. As Kwan points out, Lin Shu did have such 'political purpose in mind' when he chose to translate Haggard's works and had developed a special liking for him (Kwan 2013, p. 33). Lu Jiande, a contemporary Chinese critic, has reassessed the significance of Lin Shu's translations by arguing that Haggard's novels were 'conducive to the shaping of national identity in the target language country' and *She*, the most influential of his works, 'practically revealed the male anxiety in the Victorian era: the female authority, racial purity, and the phenomenon of regression' (Lu 2016, p. 138).

Apart from political purpose, Lin Shu also attached importance to aesthetic purpose. As Victor H. Mair, an American sinologist, notes, 'Lin Shu and his language informants' choice of subject-matter favored novels that had powerful emotional and moral appeal. Often, the translator himself was moved to tears by the story of human suffering and injustice in the works he was translating' (Mair 2010, p. 1064). By writing about 'a restricted world' with mild irony, Austen had prevented herself from being superficially received or frivolously approached. The majority of readers in a pre-modernized China would find it difficult to identify or emphasize with the protagonists of her stories. Obviously, in the eyes of the early Chinese translators, she would not be an immediate success if translated. This can be evidenced by the enthusiastic reception of her works in contemporary China, because today's readers who live in a relatively modernized society can easily appreciate the ordinariness or 'triviality' celebrated in her novels.

Gender considerations were not likely what most modern Chinese translators had intentionally incorporated in their translation choices, but gender had certainly played roles in the cross-cultural encounters. Apart from the great majority of established male authors they were obliged to be confronted with or take advantage of, they would have to address the great majority of male readers, because only few women could have access to education in early modern China. Lin Shu's translation has been denigrated as 'unfaithful' to the original texts, because his work 'epitomizes a translation process that privileges reception' (Gao 2010, p. 26). He 'chose and responded to the source texts with a view to the target culture' (Gao 2010, p. 28), so that his translated works could be readily accepted by male Chinese readers. Owing to this reception-centered strategy, Austen lost the priority of being selected in early Sino-British literary communications. As a consequence, Lin Shu has been criticized by both his contemporary and subsequent critics for wasting too much time and energy on 'many second-rate and third-rate worthless works' (Lu 2016, pp. 135–36). Generally speaking, the absence of Austen reflects the conspicuous lack of canon consciousness on the part of the early modern Chinese translators.

### 3. Depthless Knowledge of Austen

It was not until 1917 that Wei Yi (1880–1930), a major translator in the early decades of the twentieth century, made an essential introduction to Austen for the first time in *Brief Profiles of Western Famous Novelists*. He had been one of Lin Shu's major collaborators since *The Black Slaves' Plea to Heaven* in 1901, but Austen was unfortunately absent from their choices. As has been argued, this was not because modern Chinese translators demonstrated a particular disregard for female authors. As a matter of fact, many female authors had been successively introduced, translated, or enthusiastically eulogized since the emergence of the Chinese Women's Movement at the end of the nineteenth century. Austen was found to have had a notable lack of exposure among Chinese intellectuals partly due to the limited or depthless knowledge of Western literature too.

The first Chinese modern university was founded in 1898, but there had been no professional literary critics for quite a while. Literature had not become an independent academic discipline in Chinese universities until the late 1910s (Yu 2016, pp. 219–26). For the first two decades of the twentieth century, few intellectuals had made any critical or profound investigation of English literature. Their understanding of foreign authors' importance in literary history was simply acquired from reading foreign documents or overseas materials, while translators like Lin Shu had to depend upon his oral collaborators. Wei and many other translators who had received education at the Christian schools and colleges or taught themselves in literary criticism seemed to have had no better command of, or more professional approaches, to Western literature. Some others even depended on what they could find via a Japanese intermediary, because many of them had studied abroad in Japan or escaped from political persecution after the failure of the 1898 Reform Movement.

Liang Qichao (1873–1929), a leading reformist and 'a zealous advocate of political fiction' (Chi 2018, p. 30), translated four fictional works all from Japanese into Chinese in the early 1900s. They included a Japanese political novel, two French science fiction, and Allen Upward's short story 'The Ghost of the Winter Palace'. He also incorporated part of Byron's 'The Isles of Greece' in his own political novel *The Future of New China* in 1903. As an initiator of female education in Late Qing China, Liang published an essay entitled 'On Women's Learning' in 1897, in which he says, 'when I try to deduce the deepest underlying reason for the weakness of a nation, it always starts from the lack of education for women' (Liang 1999, p. 30). He attached great importance to the correlation between 'women's education' and 'the strength of the nation' (Karl and Zarro 2002, p. 184). However, having been regarded as the predecessor of the Women's Movement in modern China, Liang did not cast his eyes on any female authors in his translation, including Austen, whose works were neither 'political' in an obvious sense, nor available in Japanese before the 1920s (See Brodey and Hogan 2008).

With the awakening of women's consciousness during the movement, there had emerged a number of women's newspapers and periodicals since the end of the nineteenth century. They 'were not only effective tools for women's elementary education, but also the important front for the advocacy of women's rights' (Zeng 2021, p. 185). These new media had played a significant role in introducing and translating foreign women authors as part of the movement after the unprecedented success of Lin Shu's *The Black Slaves' Plea to Heaven*. The introductory comments on Mrs. Stowe would be a good example of insufficient and even superficial understanding of the female authors on the part of Chinese intellectuals. They wrote much about Mrs. Stowe but knew substantially very little about her as a novelist, let alone Austen.

In 1902, the year after *The Black Slaves' Plea to Heaven* was published, an article titled 'A Biography of Lady Pi Cha' attracted a wide readership among the Chinese intellectuals. 'Lady Pi Cha' was recognized as 'an outstanding heroine' and idolized as 'a female saint' (Anon 1902, p. 4), because she was thought to have changed the destiny of African-American slaves with her pen and ink. The editor emphasized in a postscript that the Chinese people had suffered as much as African-American slaves, and wished that two

hundred million Chinese women should take her as a good role model (Anon 1902, p. 4). The article was reprinted repeatedly in many women's newspapers and periodicals, such as *Women's Newspaper, Chinese Women*, *The Lingnan Women's Journal*, etc. Qiu Jin (1875–1907), a famous revolutionary feminist, wrote a poem holding Lady Pi Cha in high esteem and wishing she could be able to write a masterpiece like *The Mayflower* for the purpose of promoting women's emancipation in China. *The Mayflower*, Mrs. Stowe's collection of essays and short stories that was mentioned quite a lot in the article, was completely confused with *Uncle Tom's Cabin*. 'Pi Cha', the new Chinese translated name of Mrs. Stowe, and 'Si Tu Huo,' the one in Lin Shu's translation, were utterly misunderstood as two different American novelists.

For the decade before 1911 when the Qing Dynasty collapsed, Mrs. Stowe remained the best-known and also the most influential woman novelist among Chinese readers. In 1905, Ding Chuwo, the editor-in-chief of the periodical *The Female World*, extolled Mrs. Stowe as 'the goddess of freedom' and 'the predecessor of female revolutionaries' in his long article imbued with political comments instead of literary criticism (Ding 1905, p. 1). At the same time, many other female authors, including the best-sellers in Lin Shu's translation, found their first entries into China. In 1907, George Eliot was found to be introduced to Chinese readers probably for the first time. Qi Dan's article 'An Introduction to the English Novelist Eliot,' published in *The Chinese New Female World*, was less a discussion of George Eliot's writings than a sketch of her life as 'an outstanding woman' with a purpose to introduce 'the moral conduct of a Western woman' to 'the darkness of the Chinese female world' (Qi 1907, p. 4).

Chinese intellectuals in the Late Qing Dynasty and also in the early Republican period, including both the male and female promoters of women's rights, did not have a clear picture of Anglo-American literature and was strongly inclined to highlight the identity of 'an outstanding woman' instead of 'a successful novelist'. For the most part, their feminist intentions overwhelmed their literary purposes, which reflected a conspicuous absence of canon consciousness. As a consequence, various female novelists were accepted indiscriminately with the same underlying belief that popularity equals greatness.

For example, Zhou Shoujuan (1895–1968), one of the great translators of the early Republican period, wrote in an article that Gorge Eliot had earned widespread literary fame all over the country, and noticed that 'acclaims and praises were heaped on her like snowflakes' (Zhou 1911, p. 48). Eliot, who 'emerged unprecedentedly in the English female world', was introduced more as a successful woman than a serious novelist (Zhou 1911, p. 48). Perhaps in Zhou's eyes, she was not more important than any of the other Anglo-American female authors whose works he had translated into Chinese between 1911 and 1917, including Maria Edgeworth's *The Bracelets: A Tale*, Mrs. Gaskell's 'The Sexton's Hero', Eleanor Atkinson's *Lincoln's Love Story*, and short stories by Marie Corelli, Beatrice Grimshaw and Elinor Glyn. Most likely, it was due to George Eliot's popularity rather than her recognized canonicity that her novella *Silas Marner* was translated by Zhu Linxian, the alias of an anonymous translator. It was published in *She Bao*, the earliest and most influential newspaper in Shanghai, in 1914.

From the Women's Movement in Late Qing China to the intellectual revolution during the New Culture Movement in the mid-1910s, a number of women novelists were translated or introduced. On one hand, they were brought into modern China more by fortuity than as a product of penetrating critical insights. Many of them, for instance those translated by Lin Shu and Zhou Shoujuan, were bestselling authors or well-received writers in the Anglo-American literary world. Their works perhaps enjoyed immense popularity at the time but still needed to be ascertained as canons by means of what Samuel Johnson had called 'the test of time'.

On the other hand, Austen, the major woman novelist in the early nineteenth century, was not ignored, despite her initial absence in the high tide of literary translation. It proved inevitable that the stars of literary heaven would finally shed her bright and sparkling light upon the Chinese land. Chinese students at the Christian schools or colleges in nineteenth-

century China may have encountered Austen in their classrooms or libraries. If we take Mrs. Stowe as a contrasting example, we may understand that Austen had not been absent at all. According to Michael Gibbs Hill, 'It is likely that *Uncle Tom's Cabin* was taught at St. John's College, at least recommended to students while Wei Yi studied there' (Hill 2013, p. 58). Edie Wong also notes, 'Wei Yi most likely encountered *Uncle Tom's Cabin* while studying English at St. John's University, a missionary institution in Shanghai' (Wong 2015, p. 271). Similarly, Wei or some other Chinese students may have read or written about Austen when they were at school or college. As Shao Ying, a student at St. John's University, wrote in a student's journal, 'What our classmates read were literary works, whose author was Lady Jane Austen. She was a great novelist during the French Revolution' (Shao 1916, p. 9).

It is believed that many Chinese intellectuals could have also read about Austen in the English newspapers, especially those founded to serve foreign nationals who had stayed or settled in China since the end of the Opium War in 1842. In particular, the Chinese correspondents who were employed to report news about China would not refrain from reading the literary news or critical articles in their own newspapers. They might have read such an article titled 'Western Literature for China' in *The North-China Herald*, in which there is such a sentence: 'Before China can appreciate the distinction between Calvinism and Presbyterianism or understand Mr. Howell's preference for Jane Austen's novels over Thackeray's, she must begin at the beginning of things, and start with a course of Augustine and Boccaccio' (Anon 1890, p. 133). According to the digital data in the Shanghai Library, some other English newspapers, for example, *The China Press* and *The North-China Daily News*, published quite a few English articles exclusively on Austen, such as 'Genius of Jane Austen,' (Anon 1913, p. 5) and 'The Anniversary of Jane Austen' (Anon 1917, p. 4).

In 1917, Wei Yi, the pioneering promoter of Austen, began to eulogize her as 'one of the celebrated English novelists' for the first time in his Chinese language book *Brief Profiles* (Wei 1917, p. 24). He mentioned four of her novels to Chinese readers and regarded her as 'the founder of domestic fiction' (Wei 1917, p. 25). He put a high value on her art of characterization, and ranked her with William Shakespeare by quoting Walter Scott, Thomas Macaulay and George H. Lewes. Wei claimed that *Brief Profiles* was 'translated and edited' (Wei 1917, p. 2). In other words, it was not a result of his original research but largely a product of translating and rewriting. However, he made groundbreaking contributions to the dissemination of Austen's literary fame in China. Perhaps under the influence of Scott, he was among the earliest Chinese commentators who had taken notice of the 'narrowness' in her subject matter, which was continually referred to as a fundamental 'shortcoming' of her writings by subsequent critics.

Austen's literary light was too bright for the Chinese intellectuals to ignore, but her canonical place was not instantly recognized, and her novels remained deeply unexplored. In the 1920s and 1930s, critics made positive comments on Austen, but their judgements were reserved and contradictory. Zheng Zhenduo, a prominent critic and translator, identified Austen as a great novelist in *An Outline of Literature* (Zheng 1927, pp. 77–79), an influential monograph of its time, while, on the other hand, Jin Donglei still treated her as a minor novelist ten years later. In *An Outline of English Literary History*, the first full-length history of its kind in Chinese, Jin devoted less than half a page to Austen (Jin 1937, p. 302). In early modern China, few people had made independent studies or taken a close reading of her novels. Most critics derived their viewpoints primarily from foreign reference books available to them. For example, Zheng's basic knowledge of Austen showed distinct signs of influences by John Marcy's *The Story of the World's Literature* (Marcy 1925, pp. 355–56).

## 4. The Wartime Translators

None of Austen's novels had been translated into Chinese until 1935 when two translated versions of *Pride and Prejudice* made their appearances for the first time. One, by Dong Zhongchi, was published by the University Press in Beijing, and the other, by Yang Bin, was published by the Commercial Press in Shanghai. Although Liang Shiqiu, a

distinguished translator and literary critic, was invited to preface the former and Wu Mi, another distinguished translator and literary scholar, invited to preface the latter, both of the translations were generally unsuccessful and did not attract as adequate attention as many other translated novels. Specifically, the wartime environments in modern China had severely hampered their efforts to translate *Pride and Prejudice*. Henry Seidel Canby, an American critic, may not have known that two Chinese translators had worked on Austen's foremost masterpiece in wartime about ten years before he wrote his essay 'The War and Jane Austen' in 1942. As he says, 'The greatest novels (in English at least) written in wartime are unquestionably Jane Austen's' (Canby 1942, p. 26). However, he might not have known about the true Chinese version of 'the War and Jane Austen' in the making of a canonical author under very different historical circumstances.

Dong translated the novel when she was twenty-one years old, almost the same age as Austen was when she wrote it. In 'The Translator's Preface,' Dong mentioned that she was a college student in Beijing in 1933 when the city was seriously endangered by Japanese air bombing, since Japan occupied the north-eastern region of China in 1931. She was obliged to return to her hometown in the south, where she began to read *Pride and Prejudice* to relieve the boredom of being restricted at home and occasionally practiced her translating skills on the classic work after finding herself enjoying the story very much. She did not complete her translation until she returned to Beijing in the second year. She was not confident of her translation and felt worried about the errors and mistakes that she may have committed on account of the discontinuity and disturbance (Dong 1935, pp. 1–3). So it proved that Dong's version was not well received by either readers or literary critics, although Liang heartily recommended it in his preface. Dong was sharply criticized as 'a bad translator' by a contemporary reviewer, who believed that her translation was 'full of mistakes' because she did not show 'a full understanding of the original' (Gu 1936, p. 44).

Yang Bin, the pen name of Yang Jiwei (1905–1957),[2] was inspired to translate *Pride and Prejudice* under the influence of Wu Mi and Grace M. Boynton, two of her teachers, when she was a college student in Yenching University in 1931. Wu was one of the earliest professors who taught Austen at college and may have stimulated Yang's interests in the novel. Boynton, who taught English literature at Yanching, may have also kindled her great passion for Austen. As Xiao Qian, a former classmate of Yang's and a prominent translator of James Joyce's *Ulysses*, recalls in an article:

> Miss Boynton worked for more than thirty years in Yanching, and Yang was one of her favorite students. When she taught 'A History of English Novels,' what she was most enthusiastic about in lecturing was Jane Austen. I remember that Yang translated *Pride and Prejudice* and had it published by the Commercial Press (in Shanghai). It was the only foreign literary work that she had ever translated all her life. (Xiao 2018, p. 48)

The reason that Yang's version had not been published until four years later is probably that it took so much time for the Commercial Press to resume normal work after its headquarters had been completely destroyed during the January 28 incident in 1932 when the Japanese troops bombed Shanghai on the pretext of anti-Japanese protests.

One of Yang's great contributions to the reception of Austen in China is her translation of the English title into Chinese as '*ao man yu pian jian*' (傲慢与偏见[3]) (Yang 1935a). It has finally prevailed over half a dozen of others and created an idiomatic phrase in Chinese that is most frequently used nowadays. However, Yang's Chinese version of *Pride and Prejudice*, reprinted four times before 1949 and never after 1949, cannot be identified as 'classic' or 'influential,' because it exerted little impact upon Chinese readers. Twelve years after it was published, Professor Qian Gechuan, Dean of the Literature Department at Taiwan University, started work on Austen's foremost masterpiece without knowing that it had already been translated into Chinese, and had his translation serialized in *Peace Times*, a newspaper in Taibei. As soon as he was told by a friend the next year that there had been two Chinese versions available, he decided that he should give up his translation if he

found the novel had been well translated. He managed to acquire a copy of Yang's, but only to find that there were many errors and improprieties.

In his article 'Of the Chinese Translation of *Pride and Prejudice*,' Qian presented many examples of Yang's translation in details, based on chapter twenty, which he had just finished. By checking Yang's version in comparison with the original English novel, he revealed clearly that phrases or sentences, such as 'Lizzy shall be brought to reason . . . '; 'very desirable wife,'; 'In everything else she is as good-natured as a girl as ever lived,'; 'I have not the pleasure of understanding you,'; 'though his pride was hurt, he suffered in no other way,' were not well translated at all, and more mistakes were conspicuously found in the last two paragraphs of the chapter (Qian 1948, pp. 11–14). In 1948, Qian continued to have his translation published in instalments in *Chung Hwa English Fortnightly*, a journal established in Chongqing, the wartime capital of the Republican China during the Anti-Japanese War (1937–1945). As editor-in-chief of the magazine, he would complete his editorial job in Taibei before it was printed and distributed in Shanghai. When the People's Liberation Army took control of Shanghai in 1949, the connection between the editing and the printing discontinued. As a consequence, the magazine was closed down. Qian, a brilliant essayist and translator, quit his translation and never picked it up again.

Austen, who had a good talent for writing novels, requires an almost equivalent talent who could be qualified to engage in the job in another language, but unfortunately her novels did not capture the attention of a devoted or accomplished Chinese translator for various reasons in early modern China. Austen's literary fame would have been widespread much earlier if her foremost masterpiece had been rendered with accuracy into fluent Chinese. Successful examples that can be cited are the early translations of Shakespeare's plays by Zhu Shenghao, Chaucer's poems by Fang Chong, Milton's *Paradise Lost* by Zhu Weiji, Byron's poetry by Su Manshu, Shelley's poetry by Guo Moruo (Zha and Xie 2004, pp. 76, 147, 152, 349, 357–58). Zhu, Fang, Fu, Zhu, Su, and Guo were all accomplished translators or established men of letters. They have now been inextricably linked to those English literary giants in the history of Chinese translated literature. This may partly explain why Austen's novels had been consigned to oblivion by the general reading public as well as most Chinese intellectuals, who did not have a wide acquaintance with English literature in the pre-1949 period.

By taking Zhu Shenghao, also a wartime translator, as an individual example, we may argue that the construction of canonicity in another language needs the persevering endeavor of a devoted translator as well as a talented one. Zhu started to translate Shakespeare's plays when he was in his early twenties, almost the same age as Yang when he translated *Pride and Prejudice*. He completed translating nine plays up to 1936 when he had been an English editor at the World Book Company in Shanghai, but all his manuscripts were literally destroyed, together with a variety of notes and research materials, when he escaped from Japanese-occupied Shanghai in 1937. He resumed his translation work on Shakespeare indefatigably with an ultimate ambition to cover all the thirty-seven plays and finally completed a total of thirty-one during the war before he died in 1944. With regard to his contributions and influences, the Chinese-American literary critic Tonglin Lu has remarked that 'Zhu has made Shakespeare a part of Chinese culture through his lively translations, which are loved not only by highly educated scholars but also by ordinary amateurs of literature, including children.' (Lu 2012, p. 531).

In comparison, Yang was a talented young student, who had excellent family and middle-school education, and was admitted to Yanching, one of the few top universities in Republican China. After graduating from Yanching, she became a journalist of *Ta Kung Pao*, a well-known Hong Kong newspaper, and devoted herself to secret anti-invasion activities in the disguise of her new career in occupied Shanghai. She had chances to develop herself into a mature and skillful translator based on the fact that she had majored in English Literature at the University of Yanching, and her Chinese proficiency was substantiated by her variety of writings, including poetry, essays, novels, and short stories (Yang 1984, pp. 580–83). However, she wandered away from literary translation, with the exception

of her translation into English of Mao Tse-tung's *On Protracted War* in 1938 which was published in *Candid Comment*, an English magazine in Shanghai, with the help of the famous American writer Emily Hahn (1905–1997) and the Chinese poet and translator Shao Xunmei. She also partly translated the great Chinese novelist Shen Congwen's 'A Country Town' into English with the new title 'Old Mrs. Wang's Chickens' (Shen 1940–1941, pp. 274–80).

It is a pity that Yang never returned to her translated work of *Pride and Prejudice* for a revised or polished edition. Yang's version was printed four times before 1949, but few Chinese intellectuals or ordinary readers have ever claimed that they have been greatly moved, impressed, or influenced. Apart from mistakes and improperness, Yang's version was partly characterized by the awkwardness of her Chinese expression, lacking the fluency or eloquence with which Zhu utilized his excellent command of the modern Chinese language, and the critical sensibility with which he dealt skillfully with Shakespeare's literary canons. Provided that Yang had continued to revise her version or re-translate *Pride and Prejudice*, especially when she was a senior editor of *People's Daily* in Beijing after 1949, it was highly likely that a much better or even canonical version would have been presented to Chinese readers.

Moreover, considering her revolutionary personality and her personal struggle for freedom of choice to marry when she was young (Against her parents' wishes, she broke her pre-arranged marriage when she was very young), she might have sensed the subtle female consciousness behind 'rational' Elizabeth in Austen's foremost masterpiece. As Edgar Snow introduces about Yang's short stories in *Living China: Modern Chinese Short Stories*,

> [They are] not widely known, but she has had a strong personal influence on the development of several of China's youngest and most vigorous writers. Her courage and daring in utilizing social material heretofore tabooed in Chinese literature show an emancipation which will astound those who have persisted in believing that Chinese art is incapable of a sharp revolutionary break with the past. (Snow 1936, p. 301)

In some of the short stories she published in 1935 and 1936, she had already touched upon the gender problem in patriarchal China, such as men's superior status over women, the lack of freedom of choice to marry, women's economic independence, the conflict between a woman's job and her role as a mother, and the plight of lower-class women.[4] However, in 'A Critical Introduction to Jane Austen,' which was included as a translator's preface in her version, Yang, in spite of treating *Pride and Prejudice* as representing 'the highest achievement of domestic satire' (Yang 1935b, pp. 9–11), failed to address any gender issue in her review. Furthermore, she employed a traditional male gaze into women by first describing Austen's appearance as '*shuo ren qi qi*' (Yang 1935b, p. 10), a poetic line taken from *The Book of Songs* which means 'a splendid woman and upstanding' (Arthur 1996, p. 48), and then as '*shen cai yao tiao*' (Yang 1935b, p. 10), which means 'having a slender and sylphlike figure.' She continued to eulogize her as '*hei mei ren*' (Yang 1935b, p. 10), which literally means 'a black beauty'.

Xie Tianzhen believes that translated literature in China should be considered part of modern Chinese literature. In his opinion, the translation of literature was as important as the production of literature, and literary translation as a distinctive literary practice should not be disregarded. As he strongly argues, 'Literary translation is a form of literature production, as well as a form of existence for a work of literature. From this vantage point, literary translation and literature in translation have their own aesthetic value' (Xie 2011, p. 113). The long-time absence of 'successful' translations of Austen's novels had been an obstacle to the full recognition of her canonical place in China. It was not until 1956 that Wang Keyi produced a much more influential version of *Pride and Prejudice*. It is a great spectacle in the world that, up to now, more than sixty complete versions of the novel have been published in China, quite a few of them successfully rendered and widely recognized. All of these, canonical or not, have become part of contemporary 'translated literature,'

influenced many generations of Chinese readers and reshaped the 'global' Austen in a Chinese way.

**Funding:** This research received no external funding.

**Institutional Review Board Statement:** Not applicable.

**Informed Consent Statement:** Not applicable.

**Data Availability Statement:** Data available on request from the author.

**Conflicts of Interest:** The author declares no conflict of interest.

## Notes

[1]    Huang Mei explores the Chinese reception of Austen since 1949 with an emphasis on some particular issues between 1989 and 2009 (Huang 2012, pp. 157–65). Sun Shuo highlights the "disregard" for Austen in the early Chinese reception (Sun 2020, pp. 1–19). My essay investigates the critical reception of Austen in China for the past one hundred years (Zhang 2011, pp. 103–14).

[2]    Yang Gang was her other pen name for when she engaged in revolutionary activities later.

[3]    The Chinese title is seemingly a literal translation of the English, but both "pride" and "prejudice" could be translated into several Chinese synonyms. The former had "*ao man*" (arrogance), "*jiao ao*" (a proper sense of pride), and "*zi ao*" (self-conceitedness), while the latter had "*pian jian*" (bias), "*pian xin*" (partiality), and "*pian zhi*" (bigotedness). The combination of "*ao man*" and "*pian jian*" has been overwhelmingly accepted.

[4]    These stories include 'Corporal Punishment,' 'Sacrifice,' 'Between Father-in-law and Daughter-in-law,' 'The Suffering of a Mother,' and 'An Unofficial Biography of Huan Xiu.'

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
