# Peer review of "The Reception of Jane Austen in Early Modern China: A Canonical Perspective"

_humanities, doi:10.3390/h11040090_

Round 1
Reviewer 1 Report
Please give some more details in n. 50. I would recommend to give the full names in all cases (e.g. p. 4).
Please add some info on the 'anti-Janeites', on "The Black Slaves' Plea to Heaven" & Mrs. Stone.
Can you give some details on your findings in the digi-data of the Shanghai Libr.?
Please clearly mark the cit. on p. 9, starting in l. 376 (Miss Boynton...) and p.11, from l.435-7 (Zhu...) and p. 12, ll. 468-72.
Please list the successful examples (p. 10, l. 418), probably also in the list of references.
In the list of ref. an update of the last access to Brodey's article would be fine.
I enjoyed reading the paper. It's very clear, an interesting continuation of your research and shows a special focus not only on canonic lit., but also on a kind of "gender gap" and the importance of women's (higher) education.
Author Response
Dear Reviewer,
Thank you very much for the comments and suggestions. They’re very helpful for revising and improving my paper. I’ve studied your comments carefully and revised everything that has been requested to. All revisions made to the manuscript have been marked up. My responses to your suggestions are as follows.
- Please give some more details in n. 50. I would recommend to give the full names in all cases (e.g. p. 4).
I’ve provided details in n. 50. Full names have been given.
- Please add some info on the 'anti-Janeites', on "The Black Slaves' Plea to Heaven" & Mrs. Stone.
I’ve revised the sentences by providing some more information on all of them.
- Can you give some details on your findings in the digi-data of the Shanghai Libr.?
Yes. I’ve revised the sentence and provided two footnotes.
- Please clearly mark the cit. on p. 9, starting in l. 376 (Miss Boynton...) and p.11, from l.435-7 (Zhu...) and p. 12, ll. 468-72.
Yes. I’ve integrated the citations into my paragraphs.
- Please list the successful examples (p. 10, l. 418), probably also in the list of references.
I’ve revised the sentence and provided a footnote.
- In the list of ref. an update of the last access to Brodey's article would be fine.
Yes, the year “2012” was a mistake. It has been revised as “2022”.
With thanks and best wishes,
Yours sincerely
Helong Zhang

Reviewer 2 Report
This paper offers an interesting viewpoint and reading of Austen as a writer of fiction with appeal to readers in a range of countries, in this case, in China. The focus in this paper is really on the barriers to Austen being translated earlier and why those barriers existed. The discussion covers a range of issues regarding translating Austen’s works in China, particularly the barriers to her works even being considered for translation. The article appears well supported with references.
The paper is generally well structured and written in a clear way, though there are some opportunities to make the discussion and writing stronger in my view.
More context could be given to the discussion in the early pages. I suggest that the author consider unpacking the terms “canon” (and/or “canonization”) and some of the issues relating to “translation” in the wider scholarly research. The idea of a “global Austen” could also be explored briefly (2-3 sentences), drawing upon relevant research.
The writing is generally sound however there are some awkward phrases and aspects of the expression that can be improved:
- The author should avoid contractions (such as I’d, it’s, didn’t, hadn’t) and instead, use the full words.
- I suggest avoiding the use of “etc” in academic writing.
- Try to avoid “but” in academic writing. Consider using “however”, “yet” or a similar term.
- Always integrate quotes from sources into your writing rather than inserting them as a standalone sentence.
- Avoid starting sentences with “It”.
- Use of “the” before a proper noun needs attention throughout the paper. Sometimes “the” is inserted before a proper noun when it is not needed; other times it is missing when it is needed.
Line by line suggestions are as follows:
Abstract lines 3-4, consider indicating who has been disregarding or disapproving of Austen.
Line 18- remove “the” before Chinese
38- remove “the” before Chinese
47- sentence beginning with “It”, consider replacing “it” with what the “it” refers to. This will keep the flow of ideas clear for your reader.
Lines 47-56 “rendered” is used multiple times, consider revising.
66- sentence starts with “It”, please revise so the key idea is clear.
82-82- consider adding a sentence to explain what this means for translating texts into Chinese.
87- why “his home country”? should “his” be “their”?
92- should the footnote number be after the page number?
99- add “the” before “English literary world”.
113-115- integrate this quote better into the paragraph.
115 add “the” between “flooded” and “market”
124-125- clarify when this refers to, perhaps at a particular point in time or overall.
129- remove “ones” from the end of the sentence. It could just be “male” or “males”
129-132 consider clarifying which time period this applies to
136 “had had” is awkward, consider revising
138-142- this sentence is very long, consider revising into 2 sentences
142- remove “the” before “Chinese readers”
150 use of “we” is awkward, consider revising
152 the shift from “we” in the past sentence to “they” in the next sentence is noticeable. “They” should be reworked to clarify who exactly the “they” refers to
165-168 integrate the quote please
176-177 the first part of the sentence up to the comma is a little convoluted, consider revising so the idea is clear.
184-186 the phrase “Austen’s chances to lose her priority of being selected in the early Sino-British literary communications” is a bit wordy, consider revising.
196 consider replacing “circumstantiated” with “supported” and rework the “It” at the start of the sentence.
212 replace “aboard” with “abroad”
227 the phrase, “With the considerable increase of women’s consciousness” needs reworking and clarifying as it reads as a generalisation. All women?
232 include the author of The Black Slaves’ Plea to Heaven
286 insert “the” before “Anglo…”
287-88 replace “moment” with “time”
291 “in the high tide” is an unusual phrase, consider revising.
322 “it was not an original research” is an awkward phrase, consider rephrasing to “it was not the result of her original research” or something similar
356 insert “the” before “Translator’s place”
375 use of colon is awkward- the paragraph that starts on 376 is not clearly a quote.
384 “January 28 Incident”- clarify why Incident is capitalised and briefly state what this incident was
4052 consider adding “of the chapter” to “in the last two paragraphs” at the end of the sentence if this is what you mean.
435-438 integrate quote
442 remove “the” before “anti-invasion activities”
444-445 “competence” could be “competency”
445 “by her major study at the English department” is a little awkward, consider rephrasing
470 consider replacing “heretofore” with “previously”
479 “failed to address any gender issue”- do you mean in China specifically? Clarify please
491 “add an “s” to “translation” and “had” to “have”
497 “influencing a generation of Chinese readers after another” is awkward- do you mean “influencing many generations” or something similar?
Author Response
Dear Reviewer,
Thank you very much for the comments,corrections and suggestions. They are very valuable and helpful for improving my paper. I’ve really learned a great deal. I’ve revised everything that has been requested to. All revisions made to the manuscript have been marked up. Here is a brief summary of my responses to the suggestions.
- The author should avoid contractions (such as I’d, it’s, didn’t, hadn’t) and instead, use the full words.
All the contractions have been rewritten into full words.
- I suggest avoiding the use of “etc” in academic writing.
“etc.” has been deleted.
- Try to avoid “but” in academic writing. Consider using “however”, “yet” or a similar term.
Sentences with “but” have been rewritten.
- Always integrate quotes from sources into your writing rather than inserting them as a standalone sentence.
All the quotes have been integrated into the paragraphs.
- Avoid starting sentences with “It”.
Sentences with “It” have been rewritten.
- Use of “the” before a proper noun needs attention throughout the paper. Sometimes “the” is inserted before a proper noun when it is not needed; other times it is missing when it is needed.
“the” has been added when it is needed, and deleted when it is not.
Line by line suggestions are as follows:
- Abstract lines 3-4, consider indicating who has been disregarding or disapproving of Austen.
Yes. It has been done.
- Line 18- remove “the” before Chinese
It has been removed.
- 38- remove “the” before Chinese
It has been removed.
- 47- sentence beginning with “It”, consider replacing “it” with what the “it” refers to. This will keep the flow of ideas clear for your reader.
It has been revised.
- Lines 47-56 “rendered” is used multiple times, consider revising.
It has been revised.
- 66- sentence starts with “It”, please revise so the key idea is clear.
The sentence has been revised.
- 82-82- consider adding a sentence to explain what this means for translating texts into Chinese.
A sentence has been added.
- 87- why “his home country”? should “his” be “their”?
Yes. It is. It has been modified.
- 92- should the footnote number be after the page number?
The page number has been deleted.
- 99- add “the” before “English literary world”.
Added.
- 113-115- integrate this quote better into the paragraph.
It has been done.
- 115 add “the” between “flooded” and “market”
Added.
- 124-125- clarify when this refers to, perhaps at a particular point in time or overall.
A time period has been given in the sentence.
- 129- remove “ones” from the end of the sentence. It could just be “male” or “males”
It has been deleted.
- 129-132 consider clarifying which time period this applies to
The time period has been provided.
- 136- “had had” is awkward, consider revising
The sentence has been revised.
- 138-142- this sentence is very long, consider revising into 2 sentences
It has been revised.
- 142- remove “the” before “Chinese readers”
It has been removed.
- 150- use of “we” is awkward, consider revising
The sentence has been revised.
- 152- the shift from “we” in the past sentence to “they” in the next sentence is noticeable. “They” should be reworked to clarify who exactly the “they” refers to
The first sentence has been revised.
- 165-168 integrate the quote please
It has been integrated into the paragraph.
- 176-177 the first part of the sentence up to the comma is a little convoluted, consider revising so the idea is clear.
It has been revised.
- 184-186 the phrase “Austen’s chances to lose her priority of being selected in the early Sino-British literary communications” is a bit wordy, consider revising.
It has been revised.
- 196- consider replacing “circumstantiated” with “supported” and rework the “It” at the start of the sentence.
It has been revised.
- 212- replace “aboard” with “abroad”
It has been corrected.
- 227- the phrase, “With the considerable increase of women’s consciousness” needs reworking and clarifying as it reads as a generalisation. All women?
It has been revised.
- 232- include the author of The Black Slaves’ Plea to Heaven
The author’s information has been given when it was mentioned for the first time in the fifth paragraph.
- 286- insert “the” before “Anglo…”
Done.
- 287-88- replace “moment” with “time”
Replaced.
- 291- “in the high tide” is an unusual phrase, consider revising.
It has been revised.
- 322- “it was not an original research” is an awkward phrase, consider rephrasing to “it was not the result of her original research” or something similar
It has been corrected.
- 356- insert “the” before “Translator’s place”
Done.
- 375- use of colon is awkward- the paragraph that starts on 376 is not clearly a quote.
The sentence has been rewritten and the colon has been deleted.
- 384- “January 28 Incident”- clarify why Incident is capitalised and briefly state what this incident was
It has been revised. (… after its headquarters had been completely destroyed during the January 28 Incident in 1932 when the Japanese troops bombed Shanghai on the pretext of anti-Japanese protests. )
- 4052- consider adding “of the chapter” to “in the last two paragraphs” at the end of the sentence if this is what you mean.
It has been revised.
- 435-438- integrate quote
It has been integrated.
- 442- remove “the” before “anti-invasion activities”
It has been removed.
- 444-445- “competence” could be “competency”
The sentence has been revised.
445- “by her major study at the English department” is a little awkward, consider rephrasing
The sentence has been revised.
- 470- consider replacing “heretofore” with “previously”
It is the original word in the quote. I’m not sure whether it could be replaced.
- 479- “failed to address any gender issue”- do you mean in China specifically? Clarify please
It has been improved.
- 491 “add an “s” to “translation” and “had” to “have”
They have been done.
- 497- “influencing a generation of Chinese readers after another” is awkward- do you mean “influencing many generations” or something similar?
The sentence has been revised.
With thanks and best wishes
Yours sincerely,
Helong Zhang

Reviewer 3 Report
I found this an interesting and informative read. It feels a bit laconic; in particular, I did occasionally think you could have said more about political and cultural contexts, but perhaps that is difficult for you and I did get a clear sense of how Austen was received in China even if not always of why.
Author Response
Dear Reviewer,
Thank you very much for your comments and suggestions. I’ve been greatly encouraged by your words. Yes, what you said about political and cultural contexts is correct. As I have discussed them in my previous paper “Jane Austen’s One Hundred Years in China,” I may have tried too much to avoid repetition in this paper. Your suggestions will be very helpful for my further research. I’ve corrected a few errors in my paper this time. All revisions made to the manuscript have been marked up.
With thanks and best wishes,
Yours sincerely,
Helong Zhang
